# Nurse Navigators' Views on Patient and System Factors Associated with Navigation Needs among Women with Breast Cancer

**Sally D. Miller [1], Robin Urquhart [1,*], George Kephart [1], Yukiko Asada [1] and Tallal Younis [2]**

[1] Department of Community Health and Epidemiology, Dalhousie University, Halifax, NS B3H 1V7, Canada; sally.miller@dal.ca (S.D.M.); george.kephart@dal.ca (G.K.); yukiko.asada@dal.ca (Y.A.)

[2] Department of Medicine, Dalhousie University, Halifax, NS B3H 2Y9, Canada; tallal.younis@nshealth.ca

* Correspondence: robin.urquhart@nshealth.ca

**Abstract:** Coordinating breast cancer treatment is a complex task that can overwhelm patients and their support networks. Though the Cancer Patient Navigator (CPN) program in Nova Scotia (NS) provides professional assistance to patients, certain groups of patients may still face barriers to accessing its services. Employing interviews and a modified Delphi approach with CPN participants, this study sought to identify factors associated with the need for navigation to help better target CPN program referrals among breast cancer patients. Six CPNs were recruited directly through the CPN program manager for interviews and surveys. The CPNs identified 27 different factors, which were divided into 4 categories: sociodemographic, psychological, clinical and health systems. While these patient factors (particularly sociodemographic) are not directly modifiable, awareness of their association with the need for navigation could be used to better target patients with a high need for navigation for referral to CPN services.

**Keywords:** patient navigation; breast cancer; psychosocial oncology; nursing

## 1. Introduction

Breast cancer is the most common cancer (excluding non-melanoma skin cancers) among women worldwide, including in Canada [1]. Quality breast cancer care involves a myriad of healthcare interactions with a variety of healthcare providers. Accessing and understanding each of these interactions can be a burden for patients already struggling with the physical and emotional toll of their cancer diagnosis. In the province of Nova Scotia (NS), as in many places, cancer services are typically centralized in urban centres despite a significant proportion of the population living outside of these centres. Thus, non-urban patients may face extra geographic barriers to accessing healthcare.

To address this issue, a number of jurisdictions in Canada [2,3] and internationally [4–7] have established patient navigation programs (or programs with analogous functions) for cancer. These patient navigation programs are provided by a variety of individuals, such as lay persons, social workers, peer volunteers and nurses, though the precise roles and services provided by each does differ. Considerable evidence indicates that nurse-led navigation programs are effective in reducing patient anxiety and depression, reducing time intervals between healthcare contacts and reducing the time spent waiting to initiate treatment, among other outcomes [8]. Their effectiveness in improving patient satisfaction with treatment is mixed to positive [9–11].

The NS Cancer Patient Navigator (CPN) program is staffed by specially trained oncology nurses and oriented toward bridging barriers, especially geographic, that impede access to healthcare services [12]. The program is composed of eight navigators located in different communities around the province outside of the provincial capital of Halifax

[13]. CPNs have three major roles: psychosocial and practical, informational and coordination of care [12]. The psychosocial and practical roles involve providing emotional support and helping patients arrange the practical aspects of accessing cancer care (e.g., travel and lodging for cancer treatments, applying for low-income assistance programs, paying for some treatments and prostheses). The informational role involves activities such as reviewing diagnostic and treatment information with the patient. The coordination of care role involves communicating with the healthcare team and following up on healthcare decisions on behalf of, or in assistance to, the patient.

There are two rationales for conducting this study. First, it is possible that certain groups of patients with difficulty accessing healthcare, who thus have a great need and capacity to benefit from navigation, might not be accessing CPN services [14]. The first step in examining this concern is to determine how to identify patients with greater need for CPN services in the cancer population. Second, patient navigation programs are relatively costly interventions, so it is important to ensure that these services are targeted toward the patients with the greatest need. Limited literature exists to inform decision-makers on which sub-groups have greater need for navigation services, particularly in the Canadian setting and from a navigator's perspective. Therefore, the objective of this study was to identify patient factors associated with greater need for navigation according to the views of CPNs in NS. We found several important factors, across sociodemographic, psychological, clinical and health system classifications.

## 2. Materials and Methods

### 2.1. Study Design

This study employed a sequential mixed methods design, using both interviews and a modified Delphi survey to explore and achieve agreement on factors associated with the need for CPN services. First, we interviewed CPNs to create an initial list of factors that they perceived were associated with cancer patients' need for navigation. Next, we conducted a modified Delphi approach to establish consensus on the list of factors among CPNs, employing two rounds of an electronic survey.

### 2.2. Study Population

All eight of the CPNs in NS were invited to participate in the study through direct communication from the CPN program manager. These CPNs practice across the province of NS, covering three (Eastern, Western and Southern) of the four management zones of the Nova Scotia Health Authority (NSHA). The CPN program does not currently offer services in the Central Zone (including the provincial capital, Halifax).

### 2.3. Interviews

Semi-structured interviews were conducted by telephone. One team member (S.D.M.) conducted all interviews. Participating CPNs were asked to describe factors they thought were associated with a greater need for/use of their services, first without any prompting, then followed by asking about factors proposed a priori by the study team based on clinical expertise and a literature search of factors associated with differential clinical outcomes or healthcare use (Table 1). The interviewer and another study team member (R.U.) analyzed verbatim transcripts using framework analysis [15] to extract any relevant patient factors. Microsoft Excel 2016 (Microsoft Corporation, Redmond, WA, USA) was used to manage the data.

**Table 1.** Factors included in study interview guide.

| Factors | | |
|---|---|---|
| Age [16,17] | Responsibility for dependents [18] | Length of diagnostic interval * |
| Income level [19,20] | Comorbidity [21] | Length of treatment interval [†] |
| Social isolation [22,23] | Stage of cancer at diagnosis [20,24] | Receipt of chemotherapy [25] |

| Education level [26] | Method of cancer detection (routine screening vs, symptom-led follow-up) | Receipt of radiotherapy [25] |
|---|---|---|
| Geographic distance from resources [17,27] | Relationship with primary care physician/family doctor [28] | Experience of treatment-related toxicity |

\* Diagnostic interval is defined as the time between the first cancer-related healthcare contact and confirmation of the cancer diagnosis. [†] Treatment interval is defined as the time between confirmation of the cancer diagnosis and the initiation of treatment.

### 2.4. Delphi Survey

All of the factors raised in the interviews were included in an electronic modified two-round Delphi survey using Opinio software (ObjectPlanet Inc, Oslo, Norway) [29]. Some factors from the interviews were probed through multiple survey items (e.g., rural geography, comorbidity). In addition, two factors not included in the interviews were added to the survey for the study team's interest: being near end-of-life status and having a cancer recurrence, as these factors may signal particularly great clinical, psychosocial and practical need for navigation. The degree to which participants thought each factor was associated with need for navigation was assessed via importance ratings of 1 to 9, using the criteria developed in collaboration between the RAND Corporation and the University of California, Los Angeles (RAND/UCLA Appropriateness Criteria) [29]. Briefly, the process of summarizing participant ratings involved dividing the 9-point rating scale into three importance brackets: ratings of 1 to 3 = "not important", 4 to 6 = "uncertain" and 7 to 9 = "important." Agreement among participants was defined as at least four participants rating an item in the same importance bracket [29]. The rating process was repeated for those items that did not achieve agreement in the first round. All analyses were conducted using Microsoft Excel 2016 (Microsoft Corporation, Redmond WA, USA).

### 3. Results

Six out of the eight CPNs in NS participated in the interviews. The same six CPNs participated in the Delphi survey. Of the non-participants, one CPN cited time constraints, and the other could not be interviewed within the study timeframe. In the interviews, participants raised a total of 22 factors potentially associated with greater need for navigation (Table 2). These were grouped into four categories: sociodemographic, psychological, clinical and health system. The most frequently and lengthily discussed category was sociodemographic, within which were the factors of low income (discussed in all six interviews), low education level, low social support and greater distance to travel for healthcare services. Context-setting quotes from these discussions are presented in Table 3.

**Table 2.** Factors investigated for association with a greater need for navigation and their agreed importance, according to interviews and Delphi survey of CPNs.

| Interview Factors | Survey Factors (if Different) | Median Score | Score Range | Importance Bracket * |
|---|---|---|---|---|
| *Sociodemographic* | | | | |
| Age | Same as interview | 6 | 5–8 | Uncertain |
| Low social support | Same as interview | 9 | 6–9 | Important |
| Low education level | Same as interview | 8 | 8–9 | Important |
| Responsibility for dependents | Same as interview | 8 | 7–9 | Important |
| Newly moved to NS | Same as interview | 8 | 5–9 | Important |
| Non-English language spoken at home | Same as interview | 6.5 | 5–9 | Uncertain |
| Rural geography (greater distance from healthcare services) | Living in a community without a community-based cancer clinic or cancer centre | 8 | 1–9 | Important |

| | Greater distance from patient residence to town of significant size (~10,000 people) | 7 | 4–9 | Important |
|---|---|---|---|---|
| n/a | Greater distance from patient residence to town of significant size (~10,000 people) | 7 | 4–9 | Important |
| n/a | Greater distance from patient residence to Halifax or Sydney [†] | 7 | 5–8 | Important |
| Low income | Same as interview | 7.5 | 5–9 | Important |
| Immigrant status | Same as interview | 7 | 5–8 | Important |
| Currently in the workforce | Same as interview | 7 | 4–7 | Important |
| *Psychological* | | | | |
| Psychiatric comorbidity | Same as interview | 8.5 | 6–9 | Important |
| Sub-clinical but significant levels of anxiety or depression | Same as interview | 8 | 6–9 | Important |
| *Clinical* | | | | |
| Higher stage/risk of mortality at diagnosis (especially metastatic diagnoses) | Same as interview | 8.5 | 4–9 | Important |
| Tumour detection method [‡] | Same as interview | 5 | 3–7 | Uncertain |
| Experiencing chemotherapy-related toxicity | Same as interview | 8 | 7–9 | Important |
| Genetic/family history of cancer | Same as interview | 7.5 | 6–8 | Important |
| Having any comorbidity | Any pre-existing comorbidity | 6 | 4–9 | Uncertain |
| n/a | Multiple/chronic pre-existing comorbidities | 7.5 | 4–9 | Important |
| Receipt of chemotherapy | Same as interview | 7 | 6–9 | Important |
| n/a | Going on to experience a cancer recurrence | 7 | 5–9 | Important |
| Receipt of radiotherapy | Same as interview | 7 | 6–8 | Important |
| n/a | Near end-of-life status | 7 | 4–8 | Important |
| *Health System* | | | | |
| Longer diagnostic interval [§] | Same as interview | 8.5 | 6–9 | Important |
| Longer treatment interval [‖] | Same as interview | 8.5 | 4–9 | Important |
| No primary care provider | Same as interview | 8 | 6–9 | Important |

* Importance ratings agreed upon by survey participants were divided into three importance brackets: 1–3 = not important, 4–6 = uncertain; 7–9 = important. [†] Halifax and Sydney are the locations of the only Cancer Care Centres in the province, which are the only sites at which patients can receive certain treatments, such as radiotherapy.[‡] Two possible tumour detection methods were discussed: screening mammogram or symptom-led/diagnostic mammogram.[§] Diagnostic interval is defined as the time between first cancer-related healthcare contact and confirmation of cancer diagnosis.[‖] Treatment interval is defined as the time between confirmation of cancer diagnosis and initiation of treatment.

**Table 3.** Supporting CPN interview quotes for selected factors associated with a greater need for navigation.

| Factors | Supporting Quotes |
|---|---|
| | *Sociodemographic* |
| Low income | "The stress for them [people of low income] is, 'Oh my gosh, I have to drive to Halifax?...' or 'Medication, is it going to be expensive? I don't have a[n] [insurance] plan.'" [CPN 6]<br>"Definitely the population that can't afford it, I do find they're reaching out a lot more." [CPN 5] |
| Low education level | "They often have a meeting with the physician or are told the cancer diagnosis, but have no idea what that means... So those patients I find, there's an extreme need to go over what those physicians said" [CPN 5]<br>"Those who are working in… complex jobs and… higher education level—they understand better… and have more resources to find what they need for information." [CPN 6] |
| Low social support | "… they might not have very many… friends or family that are close by, so they need that person to connect with." [CPN 2]<br>"If you're a senior woman but you live alone… or… someone in their early thirties and you're alone, you don't have lots of friends, your family all lives out west... that certainly does make a huge difference as far as... they have no one to support them on that daily basis." [CPN 5] |
| Rural geography (greater distance from cancer centre) | "If there is extensive travel to get in to see a physician or specialist, they may opt not to have any investigations or treatment done." [CPN 5] |
| | *Psychological* |
| High patient anxiety/ subclinical psychological distress | "Those that are anxious by nature… have more requirements and… reach out to navigation or social worker or some kind of counselling support or group resource… more often." [CPN 5]<br>"Anyone with a history of anxiety or depression—I do find those people really need more one-on-one and very active follow-through." [CPN 6] |
| | *Clinical* |
| Higher stage/risk of mortality at diagnosis (especially metastatic diagnoses) | "I would definitely say that those higher stages with the worse prognosis would be contacting me more frequently." [CPN 2]<br>"…metastatic breast cancer patients [have greater navigational need] because they're going to be followed regularly by oncologists for a longer period of time" [CPN 1] |
| Experiencing chemotherapy-related toxicity | "… before, ... I'd run across the hall to the [chemotherapy] clinic, get the answer, call them [the patient] back…I'm getting better at conveying [to patients]… that the oncology clinic is who they call…" [CPN 3] *<br>"I can think of a particular breast patient… every [chemotherapy] cycle she had an issue... so... I was a support." [CPN 3] |
| | *Health System* |
| Longer diagnostic interval | "… the ones that it took longer to diagnose… they have… more concerns in general…" [CPN 2] |

* Note: For context, certain CPN offices are located in the same building as a chemotherapy clinic.

The 22 factors raised in the interviews were explored through 27 Delphi survey items (Table 2). Only four items required a second Delphi round to reach agreement on their degree of importance. These four items were "being in the workforce at diagnosis", "greater distance from patient residence to Halifax or Sydney", "immigrant status" and "receipt of radiotherapy".

After two rounds of the survey, there was agreement that 23 of 27 factors were "important", while the other 4 factors had "uncertain" importance regarding their impact on need for CPN services. The uncertain items were "age", "non-English first language", "having any pre-existing comorbidity" and "tumour detection method".

## 4. Discussion

This study sought to identify factors associated with greater need for navigation, according to CPNs themselves. We found a total of 27 unique factors spanning 4 classifications: sociodemographic, psychological, clinical and health system. Sociodemographic factors (particularly low income) were generally found to be the most emphasized in discussions. To our knowledge, only two other studies have previously examined specific factors associated with need for navigation in cancer, both conducted in the United States [30,31]. These studies both examined the question from the patient perspective, and so their findings make an interesting complement to our study. Many reported factors overlapped among the current study and those previously published, such as low education level, low income, low social support/living alone, greater travel distance to reach healthcare and higher anxiety or depression levels [30,31]. Sociodemographic factors, particularly those related to socioeconomic status and distance from healthcare, are consistently identified as being associated with barriers to accessing quality cancer care [14,31].

Indeed, the most commonly raised factors in our study (low income, low education level, low social support and greater distance to travel for healthcare services) point to the psychosocial and practical role of the CPNs being the most important of their roles. This is consistent with the existing evidence on patient navigation programs [3] and the goals of the original patient navigation program, established in 1990 in Harlem, NY [32]. Overall, there is strong evidence from our study, prior studies [3,14,30,31] and the historical circumstances of the first patient navigation program [32] that patients' sociodemographic and psychological factors influence their need for navigation, and thus should act as triggers for program referral. Further, CPN training should continue to emphasize training for the psychosocial, practical and emotional support roles that address the needs associated with these factors.

However, clinical and health system factors clearly must be considered when understanding patient needs for navigation. Interviews indicated that patients with multiple/chronic comorbidities, worse prognosis, more intense treatment regimens, no primary care provider and longer wait intervals also have important needs that may be addressed by CPNs. Particularly for patients with poorer prognoses or more intensive treatment, CPNs seem to have a vital role in educating patients on expectations, symptoms and side effects [12].

One of the most important strengths of this study is that it is the first such examination of patient factors associated with a greater need for patient navigation in Canada and from the navigator perspective. Further, the interviews allowed CPNs to raise factors without being prompted and were not restricted to a pre-specified list of factors (though such a list was used in addition to improve study comprehensiveness). An important study limitation is that we did not explore the patient perspective—healthcare providers and healthcare users have different views on how and why they use a given service, and both are important to inform program planning and referral practices. However, our results were consistent with the patient perspective reported by previous studies, which reduces the chance of having missed vital insights [30,31]. Future research should investigate whether the identified factors are generalizable to other navigation settings and jurisdictions (particularly elsewhere in Canada) with varying navigator roles and responsibilities [33], the prevalence of these factors among a general cancer population and whether a quantitative association can be observed between these factors and use of CPN services (among NS patients and elsewhere). This will allow for a more practical understanding of which factors are most relevant to target for referral to CPN in clinical practice.

Targeting patients with such factors for referral to patient navigation has already been successfully implemented in one other known jurisdiction [31].

## 5. Conclusions

From the perspective of CPNs, there are a variety of factors associated with a greater need for navigation that may be useful in identifying cancer patients who should be targeted for CPN referral. Sociodemographic factors appear to be particularly important, and while they are not directly modifiable, they have great potential for identifying patients who should be targeted for a CPN referral.

**Author Contributions:** Conceptualization, methodology and validation, S.D.M., R.U., G.K., Y.A. and T.Y.; investigation, S.D.M.; formal analysis, S.D.M. and R.U.; writing—original draft preparation, S.D.M.; writing—review and editing, S.D.M., R.U., G.K., Y.A. and T.Y.; supervision, R.U.; funding acquisition, S.D.M. All authors have read and agreed to the published version of the manuscript.

**Funding:** This research was funded in part by the Maritime SPOR SUPPORT Unit and the Beatrice Hunter Cancer Research Institute (with support from the QEII Health Sciences Centre Foundation and the Breast Cancer Society of Canada).

**Institutional Review Board Statement:** Ethics approval was obtained through the Nova Scotia Health Authority (NSHA) Research Ethics Board (File No.: 1023809).

**Informed Consent Statement:** Written informed consent to participation was obtained from all participants involved in the study.

**Data Availability Statement:** Data are available on request due to privacy restrictions.

**Acknowledgments:** The authors wish to thank the Nova Scotia Cancer Patient Navigation program for their generous cooperation with this study and their tireless efforts to support cancer patients across the province.

**Conflicts of Interest:** The authors declare no conflict of interest.

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
