# Peer review of "Nurse Navigators’ Views on Patient and System Factors Associated with Navigation Needs among Women with Breast Cancer"

_curroncol, doi:10.3390/curroncol28030195_

Round 1

Reviewer 1 Report

I appreciated this paper and enjoyed reading it. Although the study includes only 6 people, the results are of interest to practitioners and researchers alike. PN for cancer screening, diagnostics and care is an invaluable service to support achieve various cancer outcomes (screening completion, timely diagnostic care). I have a few comments related to content and then several minor editorial comments.

Comments related to Content:

  • It would helpful to mention outcomes that can be achieved through patient navigation - e.g., improved diagnostic completion, improved timeliness of treatment initiation, improved completeness of cancer care. Adding this information would better substantiate the importance of navigation.
  • CPN is a relatively expensive intervention and, therefore, studies like this that are then translated to assessment tools can help practitioners prioritize the service for those that will most benefit (efficiency).
  • Please describe the typical training of CPNs (e.g., nurses, lay workers)
  • The results can also inform necessary training for CPNs. If psychological and sociodemographic factors are particularly important, what does this imply for training of CPNs.

Editorial comments:

  • line 10, spell out NS
  • line 12, add who interviews were conducted with, "Employing interviews a modified Delphi survey approach with CPNs..."
  • In the Introduction study, the authors use a lot of ( ). I would use commas, or other restructure the sentences to use these less frequently.
  • line 36/37: Would revise language in () to, "(e.g., travel and lodging for cancer treatments,...) and lose the "etc." at end.
  • line 64, spell out NSHA
  • line 70, were the "pre-hypothesized factors" informed by the literature? If so, please add citations. 
  • line 71, were transcripts verbatim?
  • line 86, should RAND/UCLA be spelled out?
  • Table 2 - not sure why there is a "n/a" in some places where there were not interview factors and not in others
  • line 143, you say that "there is strong evidence that patients' sociodemographic factors influence their need for navigation" - but no citations are included. Please add appropriate references.
  •  

Author Response

Point 1: It would helpful to mention outcomes that can be achieved through patient navigation - e.g., improved diagnostic completion, improved timeliness of treatment initiation, improved completeness of cancer care. Adding this information would better substantiate the importance of navigation.

Response 1: Thank you for this suggestion. We have added the statement: “Patient navigation can be provided by a variety of individuals, such as lay persons, social workers, peer volunteers and nurses. Patient navigation has been found to be effective in reducing patient anxiety and depression, treatment satisfaction, screening adherence and uptake, timeliness of diagnostic resolution and treatment initiation” with supporting citations (lines 32-35).

Point 2: CPN is a relatively expensive intervention and, therefore, studies like this that are then translated to assessment tools can help practitioners prioritize the service for those that will most benefit (efficiency).

Response 2: We agree entirely. We have added the statement: “Second, patient navigation programs are relatively costly interventions and so it is important to ensure that these services are targeted toward the patients with the greatest need” (lines 54-56).

Point 3: Please describe the typical training of CPNs (e.g., nurses, lay workers)

Response 3: We have added the statement: “Patient navigation can be provided by a variety of individuals, such as lay persons, social workers, peer volunteers and nurses” with supporting citations (lines 31-32).

Point 4: The results can also inform necessary training for CPNs. If psychological and sociodemographic factors are particularly important, what does this imply for training of CPNs.

Response 4: Thank you for this comment. We have added a statement in Discussion section: “Further, CPN training should continue to emphasize training for these psychosocial, practical, and emotional support roles.” (lines 170-171).

Point 5: line 10, spell out NS

Response 5: We have added the full spelling: “Nova Scotia (NS)” (line 10).

Point 6: line 12, add who interviews were conducted with, "Employing interviews a modified Delphi survey approach with CPNs..."

Response 6: We have added as suggested “with CPNs” (line 12).

Point 7: In the Introduction study, the authors use a lot of ( ). I would use commas, or other restructure the sentences to use these less frequently.

Response 7: We have replaced several sets of parentheses with commas (lines 37-38, 39-40, 47-48).

Point 8: line 36/37: Would revise language in () to, "(e.g., travel and lodging for cancer treatments,...) and lose the "etc." at end.

Response 8: We have added “e.g.” and removed “etc.” as suggested (lines 43-44).

Point 9: line 64, spell out NSHA

Response 9: We have added the full spelling: “Nova Scotia Health Authority (NSHA)” (line 74).

Point 10: line 70, were the "pre-hypothesized factors" informed by the literature? If so, please add citations.

Response 10: We added the statement: “pre-hypothesized factors generated by the study team based on clinical expertise and a literature search of factors associated with differential clinical outcomes or healthcare use” (lines 81-82) and added supporting citations to Table 1.

Point 11: line 71, were transcripts verbatim?

Response 11: We added detail that transcripts were verbatim: “…analyzed verbatim transcripts using framework analysis…” (line 83).

Point 12: line 86, should RAND/UCLA be spelled out?

Response 12: We added text to clarify these abbreviations. “RAND”, as in RAND Corporation, is not an abbreviation; UCLA has been expanded to “University of California, Los Angeles” (line 101-102).

Point 13: Table 2 - not sure why there is a "n/a" in some places where there were not interview factors and not in others

Response 13: We used “n/a” initially to denote instances where the survey factor had not been included in the interview any form conceptually similar, whereas the empty rows were survey factors that added more detail to the same concept included in the interview. However, we acknowledge that this may be confusing to external readers and have amended to have “n/a” in any instance where there is no interview factor that directly matches to a survey factor.

Point 14: line 143, you say that "there is strong evidence that patients' sociodemographic factors influence their need for navigation" - but no citations are included. Please add appropriate references.

Response 14: We have amended the sentence to read: “Overall, there is strong evidence from our study, prior studies and the historical circumstances of the first patient navigation program that patients’ sociodemographic factors influence…” with supporting citations (lines 167-168).

Reviewer 2 Report

The authors pose an important question and developed a sound strategy to address it.

However, the manuscript could be improved with a more reader-friendly display of the results in tables 1 and 2, as well as expanded thought in the discussion section.

For example:

In Table 1, it would be helpful to define ‘length of diagnostic interval’ and ‘length of treatment interval’ in the table notes, so that readers don’t have to look elsewhere for the information.

In Table 2:

  • More precise headers would be helpful to readers; e.g., ‘Median score’, ‘Score Range’, and ‘Overall Importance Rating’.
  • The scoring system should be explained in the table notes, not just in the methods section.
  • For the 3 survey factors under ‘geography’, list the items in rank order of importance score.
  • It’s not clear from the text why only one of 4 items of ‘uncertain’ importance (any pre-existing co-morbidity) is listed in the table.

In the discussion section, there is no mention of how these findings can or will be applied in practice. With 24 different factors identified as important, it seems likely that the majority of patients in the 3 zones of interest would have at least 1 of these factors. Can the authors estimate this percentage of patients? If it is a very high percentage, it might be more useful to provide broad education about the availability of navigation, along with tailored information about how it can help individual patients (i.e., personalized information based on a patient’s specific risk factors)

For future research, perhaps the most important research question would be to assess how valuable navigation is to patients with the identified risk factors (e.g., satisfaction with care, adherence to treatment regimens, reduction in stress/anxiety).

Author Response

Point 1: In Table 1, it would be helpful to define ‘length of diagnostic interval’ and ‘length of treatment interval’ in the table notes, so that readers don’t have to look elsewhere for the information.

Response 1: Thank you for this comment. We have added footnotes defining diagnostic interval and treatment interval in Table 1 footnote: “Diagnostic interval is defined as the time between first cancer-related healthcare contact and confirmation of cancer diagnosis”; “Treatment interval is defined as the time between confirmation of cancer diagnosis and initiation of treatment” (lines 88-91).

Point 2: In Table 2: More precise headers would be helpful to readers; e.g., ‘Median score’, ‘Score Range’, and ‘Overall Importance Rating’.

Response 2: We have renamed Table 2 column titles as follows: “Median” changed to “Median Score”, “Range” changed to “Score Range” and “Importance” changed to “Importance Bracket”.

Point 3: In Table 2: The scoring system should be explained in the table notes, not just in the methods section.

Response 3: We have added a footnote to explain the Importance Bracket column: “Importance ratings agreed upon by survey participants were divided into three importance brackets: 1-3 = Not Important, 4-6 = Uncertain; 7-9 = Important” (lines 125-126).

Point 4: In Table 2: For the 3 survey factors under ‘geography’, list the items in rank order of importance score.

Response 4: We have re-ordered the three geography survey factors so that “Living in a community without a community-based cancer clinic or cancer-centre” (median score 8, range 1-9) is at the top, followed by “Greater distance from patient residence to town of significant size (~10,000 people)” (median score 7, range 4-9) and “Greater distance from patient residence to Halifax or Sydney” (median score 7, range 5-8).

Point 5: In Table 2: It’s not clear from the text why only one of 4 items of ‘uncertain’ importance (any pre-existing co-morbidity) is listed in the table.

Response 5: Originally, the table was intended to show only factors that had been agreed by CPNs in the survey to be important; the one “uncertain” factor included in the table (“Any pre-existing comorbidity”) was included for context because another related factor (“Multiple/chronic pre-existing comorbodit(ies)”) had been considered important. However, we agree that it makes more sense to present all importance ratings, regardless of importance, for completeness and full transparency of results. Table has been amended to include the three “uncertain” factors originally left out (“age”, “non-English first language”, “having and “tumour detection method).

Point 6: In the discussion section, there is no mention of how these findings can or will be applied in practice. With 24 different factors identified as important, it seems likely that the majority of patients in the 3 zones of interest would have at least 1 of these factors. Can the authors estimate this percentage of patients? If it is a very high percentage, it might be more useful to provide broad education about the availability of navigation, along with tailored information about how it can help individual patients (i.e., personalized information based on a patient’s specific risk factors)

Response 6: We thank the Reviewer for this comment and agree entirely. We have added that the prevalence of these factors should be investigated in future research, along with the statement “This will allow for a more practical understanding of which factors are most relevant to target for referral to CPN in clinical practice” to emphasize the remaining work still required to translate our results into useful clinical practice (lines 189-193).

Point 7: For future research, perhaps the most important research question would be to assess how valuable navigation is to patients with the identified risk factors (e.g., satisfaction with care, adherence to treatment regimens, reduction in stress/anxiety).

Response 7: This is a great point, and one that has been explored in the literature in the past. In response to a similar comment from the other Reviewer, we have added some information in the Introduction section describing the known effectiveness of patient navigation, including its positive impact on patient satisfaction and patient anxiety and depression. “Patient navigation can be provided by a variety of individuals, such as lay persons, social workers, peer volunteers and nurses. Patient navigation has been found to be effective in reducing patient anxiety and depression, treatment satisfaction, screening adherence and uptake, timeliness of diagnostic resolution and treatment initiation.” (lines 31-35)

Round 2

Reviewer 2 Report

The presentation of the data in the tables is much improved.